# Three-Dimensional Focusing Measurement Method for Confocal Microscopy Based on Liquid Crystal Spatial Light Modulator

**DOI:** 10.3390/s25082620

**Published:** 2025-04-21

**Authors:** Yupeng Li, Yifan Li

**Affiliations:** College of Metrology Measurement and Instrument, China Jiliang University, Hangzhou 310018, China; p22020854061@cjlu.edu.cn

**Keywords:** liquid crystal spatial light modulator, confocal microscopy, three-dimensional focusing technology

## Abstract

Micro-nano measurement represents a critical engineering focus in the advancement of micro-nano fabrication technologies. Exploring advanced micro-nano measurement methods is a key direction for driving progress in micro-nano manufacturing. This study proposes a confocal measurement method utilizing a liquid crystal spatial light modulator (LC-SLM) to simulate a binary Fresnel lens for 3D focusing, enabling the non-mechanical measurement of spatial positions on sample surfaces. Specifically, it introduces a 3D focusing method based on LC-SLM, constructs a confocal microscopy 3D focusing system, and conducts lateral focusing experiments and axial focusing experiments. Experimental results demonstrate that the system can freely adjust lateral focusing positions. Within an axial focusing range of 900 μm, it achieves axial measurement accuracy exceeding 1 μm, with a maximum resolution capability of approximately 16.667 nm. Compared to similar confocal microscopy systems, this method allows rapid adjustment of lateral focusing positions without regenerating phase grayscale maps, achieves comparable axial measurement accuracy, and enhances measurement speed.

## 1. Introduction

With the continuous advancement of micro-nano fabrication technologies, device dimensions have been scaled down to the nanoscale. Miniaturized components are now widely applied in fields such as engineering materials [1], biomedicine [2], and microelectronics [3]. To accurately evaluate the manufacturing quality of precision-engineered devices, metrology technologies must evolve synergistically with fabrication processes, serving as an indispensable component for advancing precision manufacturing and enhancing the performance of micro-nano devices [4]. Conventional surface measurement methods include contact and non-contact approaches. Contact-based methods risk surface damage and are susceptible to environmental interference, rendering them unsuitable for rapid, precise, and non-destructive inspections. In contrast, photoelectric non-contact measurement techniques, which avoid physical interaction with surfaces, have gained prominence in microstructure characterization. Among these, confocal microscopy has emerged as a research focus in optical non-contact metrology due to its unique optical sectioning capability and exceptional lateral resolution [5].

In confocal microscopic measurements, the Nipkow disk scanning method represents a classical beam scanning approach. This technique incorporates a Nipkow disk into the optical path, featuring thousands of spirally arranged pinholes on its surface. During rotation, these pinholes sequentially scan different positions of the sample to achieve point-by-point illumination [6]. Petran et al. [7] introduced Nipkow disk rotational scanning imaging into confocal measurements to replace conventional sample or objective lens scanning. Grant et al. [8] developed a high-speed fluorescence confocal microscopy system based on Nipkow disk scanning, achieving an imaging speed of 10 frames/s. However, this method imposes stringent requirements on pinhole positioning and dimensions, involves complex fabrication processes, suffers from extremely low light utilization efficiency (resulting in weak signal intensity), and introduces mechanical errors due to its intricate moving components. The emergence of galvanometer scanners has provided a new technical pathway to enhance the measurement speed of confocal systems, with commercial confocal microscopes universally adopting galvanometers as core components [9]. Chen et al. [10] integrated galvanometer scanning with chromatic confocal microscopy to design a full-field chromatic confocal microscope, achieving a measurement speed of 8000 lines/s under extreme conditions. Beam scanning via microlens arrays is another established method. A microlens array comprises lenses with diameters ranging from tens of micrometers to several millimeters arranged in specific patterns, enabling customized performance through adjustments in lens geometry, focal length, and spatial distribution [11]. To address the slow measurement speed of single-beam systems, multi-beam parallel confocal techniques have been developed. Eisner incorporated combined microlens and pinhole arrays into confocal systems to achieve synchronous cross-sectional measurements [12]. Although microlens arrays exhibit higher light utilization efficiency and simpler operation compared to conjugate pinhole disks, their complex manufacturing processes and inconsistent focal lengths limit their applicability in high-precision metrology.

To address the limited versatility of microlens arrays, spatial light modulators (SLMs) have been increasingly adopted in confocal microscopy. These devices replace conjugate pinholes with structured light modulation modules, eliminating the need for precise pinhole alignment while enabling rapid scanning of microsurface features [13]. As a novel fully digital planar display technology, the digital micromirror device (DMD) offers advantages including fast switching speed, high brightness, superior contrast ratio, and robust reliability [14]. Nayar et al. [15,16] developed a programmable imaging system by implementing spatiotemporal control over DMD micromirrors, enabling dynamic adjustment of geometric characteristics, field of view, and resolution, with a 20 dB enhancement in dynamic range. In 2005, Ri et al. [17] pioneered the concept of a “DMD Camera”, integrating phase-shifting techniques to achieve single-image 3D reconstruction. This system implemented distinct dynamic ranges across the RGB channels of the camera, achieving a fivefold expansion in overall detectable dynamic range. Ankit et al. [18] incorporated DMD into an optical relay system utilizing diffractive gratings to split light into dual beams, demonstrating applications in adaptive color correction, wafer inspection, and high dynamic range imaging. Despite its widespread adoption, DMD technology faces limitations such as reduced optical aperture due to inter-mirror gaps, inherent mechanical and circuit complexity introducing systemic errors, and prohibitively high costs for high-resolution DMD chips, significantly increasing overall implementation expenses.

As a type of spatial light modulator (SLM), the liquid crystal spatial light modulator (LC-SLM) has been developed for more than 40 years. Although there are already devices with high resolution and fine pixel size on the market, it has the advantages of simple fabrication, low cost, and easy controllability, and has been widely used in fields such as holographic video projection and optical communication [19]. This is achieved by applying a voltage to induce the molecular reorientation of liquid crystals [20], enabling most functionalities of the DMD. Leveraging pixel-level voltage control precision, LC-SLM-modulated structured light demonstrates significant potential for high-precision measurement applications, driving growing interest in LC-SLM-based structured light modulation technologies [21]. Das et al. [22] implemented lateral 2D scanning in confocal microscopy using binary-phase LC-SLM-based beam steering, generating optical sectioning images via dynamic binary holographic scanning. The same team further developed a point-scanning microscope employing the LC-SLM, where illumination beams are adaptively tailored to sample requirements with pixel-level intensity control through LabVIEW programming. Scanning experiments on metallic coins demonstrated enhanced signal-to-noise ratios for specimens with varying reflectivity [23]. Chansuk et al. [24] proposed a time-domain low-coherence optical diffraction tomography technique based on ferroelectric liquid crystal SLM (FLC-SLM), enabling rapid angular scanning while maintaining wide-field coherence in a compact setup. These developments confirm that the LC-SLM enables lateral scanning while preserving the axial high-resolution advantages of confocal systems. However, conventional confocal microscopy remains constrained by axial defocus detection methods, necessitating time-consuming tomographic scanning during measurements. Sample stage movement is often unavoidable, resulting in low measurement efficiency and susceptibility to mechanical errors.

Based on the above issues, this paper proposes a measurement method that utilizes an LC-SLM to simulate a binary Fresnel lens for achieving changes in 3D focal positions. By digitizing the Fresnel lens and loading it onto the LC-SLM, the LC-SLM can mimic the characteristics of a binary Fresnel lens, enabling adjustable focusing capabilities. By directly controlling the position of the binary Fresnel lens, the system’s lateral focal position can be adjusted while establishing a relationship between light intensity signals and axial distances to directly obtain axial position information. Compared to traditional confocal microscopy systems, this method allows for the rapid adjustment of the lateral focal position without regenerating phase grayscale maps, thereby achieving similar precision in axial measurements and improving measurement speed.

## 2. Experimental Materials and Fabrication Methods

This chapter primarily analyzes the modulation characteristics of the LC-SLM used in the experiment and proposes a method for constructing binary Fresnel lenses.

### 2.1. Study of LC-SLM Modulation Characteristics

The liquid crystal molecules in LC-SLM are arranged in an average orientation parallel to the substrate without the application of an electric field, which ensures the stability of the liquid crystal in the controlled state and effectively regulates the propagation characteristics of light. Applying a relatively low voltage in the order of 1–10 V will cause liquid crystal molecules to realign and change the effective refractive index of light polarized along the direction of stationary molecule orientation [25]. The maximum achievable phase change is proportional to the thickness of the liquid crystal layer. Therefore, the optical phase delay introduced by wedge-shaped prisms in the aperture can be approximated by a series of stepped tilted phase delays [26].

If the LC-SLM is assumed to be an ideal structure, then its simulated wedge prism phase distribution function and far-field complex amplitude distribution function are as follows:(1)$$\varphi^{ideal}\left(\theta ,x\right)=\frac{2\pi }{\lambda }x\sin\theta$$(2)$$T\left(f_{x}\right)=D\sin c\left[D\left(f_{x}-\frac{\sin\theta }{\lambda }\right)\right]$$
where *x* is the spatial coordinate on the LC-SLM surface, *λ* is the wavelength, *θ* is the deflection angle, and *D* is the aperture of the LC-SLM. The phase delay at the thickest point of the prism (i.e., when *x* = *D*) can be expressed as follows:(3)$$\varphi_{D}=\frac{2\pi }{\lambda }D\sin\theta$$

When simulating a wedge prism with an LC-SLM, increasing the deflection angle simultaneously enlarges the wedge angle of the prism and reduces the number of pixels per modulation period. Consequently, at the maximum deflection angle, the pixel count per period reaches its minimum. Since a single period must contain at least two pixels to represent phase delays of 0 or 2π, this constraint determines the maximum achievable deflection angle.(4)$$\theta_{\max}=\arcsin\left(\frac{\lambda }{2d}\right)$$

As derived from the above equation, the maximum deflection angle *θ*_max_ is dependent on the pixel pitch *d* and wavelength *λ*. The LC-SLM employed in this study has a pixel pitch of *d* = 6.35 μm and operates with an input light wavelength of *λ* = 635 nm. Substituting these parameters into the formula yields a maximum deflection angle *θ*_max_ = 0.05 rad ≈ 2.866°.

Beam deflection efficiency refers to the ratio of light intensity deflected to the target angle to the incident light intensity during beam steering. However, the LC-SLM inherently lacks an ideal continuous structure due to its discrete pixelated architecture. Additionally, phase flyback regions caused by electric field fringing effects and nonlinear phase delay distributions further degrade the deflection efficiency of LC-SLM [27]. For analytical simplicity, under the assumption of disregarding 2π-periodic phase resets, the phase delay distribution required for beam deflection using a prism model can be expressed as follows:(5)$$\varphi_{j}^{ideal}\left(\theta ,x_{j}\right)=\frac{2\pi }{\lambda }x_{j}\sin\theta$$
where $x_{j}$ represents the position of the *j*-th pixel on the LC-SLM, with *j* = 1, 2, 3, …, *N*, and *N* denotes the total number of pixels in the LC-SLM. The transmittance function can then be expressed as follows:(6)$$t\left(x\right)=\sum_{n=0}^{N-1}\delta \left(x-nd\right)⊗\left[\exp\left(j2\pi f_{\theta }x\right)\mathrm{rect}\left(\frac{x}{d}\right)\right]$$(7)$$f_{\theta }=\frac{\sin\left(\theta \right)}{\lambda }$$
where *λ* is the wavelength of the incident light, and *d* is the pixel pitch. When the incident light is perpendicularly illuminated, the far-field distribution can be expressed as follows:(8)$$T\left(f_{x}\right)=d\sin c\left(df_{x}\right)\times \frac{\sin\left[N\pi d\left(f_{x}-f_{\theta }\right)\right]}{\sin\left[\pi d\left(f_{x}-f_{\theta }\right)\right]}$$
where $f_{x}=xf/\lambda$, N is the number of pixels, and f is the focal length of the Fourier lens. When using the prism phase distribution function, theoretically, beam deflection at any angle within a specific range can be achieved. From Equation (8), the calculation method for deflection efficiency *η* can be derived and expressed as follows:(9)$$\eta ={\mathrm{sinc}}^{2}\left[\frac{d\sin\left(\theta \right)}{\lambda }\right]$$

From this equation, the beam deflection efficiency curve for the LC-SLM used in this study can be derived, as shown in Figure 1.

From this, it can be concluded that continuous beam deflection using an LC-SLM to simulate a wedge prism is feasible. However, the deflection efficiency decreases as the deflection angle increases, and the presence of flyback regions further reduces the efficiency.

Under ideal conditions, the phase delay is continuous. However, in practical applications, the number of phase steps simulated using the LC-SLM is limited. Let *M* represent the number of equally spaced phase steps simulated using the LC-SLM. Then, the phase delay for beam deflection can be expressed as follows:(10)$$\varphi_{j}\left(\theta \right)=\mathrm{round}\left(\varphi_{j}^{ideal}\left(\theta \right)\frac{M}{2\pi }\right)\frac{2\pi }{M}$$

The normalized error for beam deflection is defined as follows:(11)$$\varepsilon_{norm}=\frac{\theta_{r}-\theta_{i}}{\theta_{spot}}$$
where *θ_r_* is the actual deflection angle, *θ_i_* is the theoretical deflection angle, and *θ_spot_* represents the far-field divergence angle or spot size of the beam. The beam deflection accuracy can be analyzed using Equation (11) [28]. The far-field divergence angle *θ_spot_* can be expressed as follows:(12)$$\theta_{spot}=\frac{2\lambda }{D}$$
where *D* is the effective working aperture of the LC-SLM. For discretely quantized step-like phase delays, the average slope of the phase delay can be obtained by fitting the step-like phase profile, and the corresponding deflection angle can be determined. This method is fundamentally consistent with the deflection angle derived from scalar diffraction theory [29]. Let *k_r_* represent the phase delay slope obtained through the average slope fitting method. Then, the actual beam deflection angle *θ_r_* can be expressed as follows:(13)$$\theta_{r}=\arcsin\left(\frac{\lambda k_{r}}{2\pi d}\right)$$

When the number of phase steps is *M* = 32 and *N* = 32, the simulation curve of the actual beam deflection angle within the deflection range of the LC-SLM is shown in Figure 2.

It can be observed that when the deflection angle approaches its maximum value, the actual deflection angle deviates significantly from the theoretical deflection angle, resulting in a larger normalized error. The LC-SLM used in this study has an 8-bit color depth, theoretically allowing 256 grayscale levels when displaying grayscale images. This corresponds to an equidistant phase step count of *M* = 256 within the phase modulation range from 0 to 2π. Since the normalized error near the maximum deflection angle is significantly larger than that at other angles, which affects the overall clarity of the error distribution curve, data near the maximum deflection angle were excluded during analysis. Data within the deflection angle range from 0 to 0.045 rad were retained for analysis, and the resulting error distribution is shown in Figure 3.

As shown in the figure, the absolute− value of the normalized error $\left|\varepsilon_{norm}\right|$ for LC-SLM beam deflection in practical applications is consistently less than 0.002%. Therefore, the absolute value of the beam deflection error $\left|\theta_{error}\right|$ satisfies $\left|\theta_{error}\right|<2\times 10^{-5}\theta_{spot}$. Based on calculations, the following can be derived as follows:(14)$$\begin{array}{cc} \left|\theta_{error}\right| & <2\times 10^{-5}\times 1.5625\times 10^{-4}\mathrm{rad}\\ & =3.125\times 10^{-9}\mathrm{rad}\approx 1.79\times 10^{-7}{}^{\circ} \end{array}$$

Through the above analysis, achieving higher beam deflection accuracy requires the LC-SLM to support a sufficiently large number of equidistant phase steps. When *M* = 256, the beam deflection error of the LC-SLM is already less than 1.79 × 10^−7^°. Therefore, it can be concluded that the actual beam deflection angle is effectively identical to the theoretical deflection angle, and the impact of beam deflection errors can be neglected in subsequent experimental processes.

### 2.2. Binary Fresnel Lens Construction Method

Traditional Fresnel lenses typically rely on complex photolithography or etching techniques for manufacturing and are difficult to achieve dynamic changes, while LC-SLM-based liquid crystal Fresnel lenses have a simple generation process, can respond quickly at low voltages, and have good transmission performance for linearly polarized light, making them suitable for various optical applications [30]. A liquid crystal Fresnel lens converges the beam at its focal point. When its focal length is adjusted, the spatial position of the convergence point also changes. This process can be equivalently described as the lens undergoing random motion within a two-dimensional plane perpendicular to the optical axis. Since the reflective surface of the LC-SLM is planar, the curved surface of the lens can be neglected, allowing for an analysis in a two-dimensional space as follows:

Let the incident light be a vertically incident parallel beam, which is reflected by the LC-SLM. Let *OF* represent the optical axis of the simulated lens, *A* be an arbitrary point on the LC-SLM, *F* be the focal point of the simulated lens, *f* be the focal length of the simulated lens, and *l* be the distance between *A* and the focal point *F*. Then, the distance *h* from *A* to the optical axis is obtained as follows:(15)$$h=\sqrt{\Delta l^{2}+2\Delta l\cdot f},\Delta l=l-f$$

If interference reinforcement is to be satisfied, it is necessary to make ∆*l* an integer multiple of *λ*, and the optical path difference ∆*l* is much smaller than the focal length *f*. Therefore, Equation (15) can be rewritten as follows:(16)$$h=\sqrt{2\Delta l\cdot f}$$

It can be considered that ∆*l* = *σλ*, where *σ* = 1, 2, 3, …, and *λ* is the wavelength of the incident light. According to the relationship between the optical path difference and the phase, it is known that each *σ* corresponds to a 2π period. Then, the number of pixels *P_σ_* contained in each period is as follows:(17)$$\begin{array}{cc} P_{\sigma } & =\frac{\left(h_{\sigma }-h_{\sigma -1}\right)}{d}=\frac{\sqrt{2\Delta l_{\sigma }\cdot f}-\sqrt{2\Delta l_{\sigma -1}\cdot f}}{d}\\ & =\frac{\sqrt{2\sigma \cdot \lambda \cdot f}-\sqrt{2\left(\sigma -1\right)\cdot \lambda \cdot f}}{d}=\frac{\sqrt{2\lambda f}\cdot \left(\sqrt{\sigma }-\sqrt{\sigma -1}\right)}{d} \end{array}$$

Among them, *d* is the pixel pitch. Set *λ* = 635 nm, *d* = 6.35 μm. As the period increases, the number of pixels in each period will decrease rapidly and then tend to be flat; that is, the width of each zone-plate period will decrease rapidly at first and then gradually become flat. Therefore, the phase delay generated by the *j*-th pixel in the plane (with the center position of the LC-SLM as the origin) for the vertically incident parallel light can be expressed as follows:(18)$$\varphi \left(j\right)=-\frac{2\pi }{\lambda }\Delta l=-\frac{2\pi }{\lambda }\cdot \frac{h^{2}}{2f}=-\frac{\pi h^{2}}{\lambda f}$$

Now expand the 2D plane to 3D space. Let the coordinates of point *A* on the LC-SLM be (*x*, *y*), where *x* and *y* are the coordinates with the center of the simulated lens as the system origin. Then, at this time, $h=\sqrt{x^{2}+y^{2}}$; therefore, the phase distribution function *φ*(*j*) can be written as follows:(19)$$\varphi \left(x,y\right)=-\frac{\pi }{\lambda f}\left(x^{2}+y^{2}\right)$$

From the structure of a binary Fresnel lens, it is evident that the zones of the Fresnel lens become narrower and more densely packed as the radius increases. If the resolution of the LC-SLM is insufficient to achieve a one-to-one correspondence between pixels and the densely packed zones in this region, the desired phase modulation cannot be achieved, and the simulation of a binary Fresnel lens with the theoretical focal length will fail. Considering the 2D discrete pixel structure of the LC-SLM, the phase gradient between adjacent pixels at the outermost region can be expressed as follows:(20)$${\left|\nabla \varphi \left(x,y\right)\right|}_{\max}=\frac{2\pi {\left(x,y\right)}_{\max}}{\lambda \left|f\right|}$$

The phase difference between adjacent pixels should be less than π; therefore, in order to obtain the expected phase modulation, it is necessary to satisfy the following equation:(21)$${\left|\nabla \varphi \left(x,y\right)\right|}_{\max}<\frac{\pi }{d}$$

From Equations (20) and (21), the minimum focal length of the simulated binary Fresnel lens can be derived as follows:(22)$$\left|f\right|>\frac{2{\left(x,y\right)}_{\max}d}{\lambda }=\frac{Nd^{2}}{\lambda }\approx 81.28\mathrm{mm}$$

Since the phase modulation depth of the pure-phase LC-SLM is 2π, its phase function should be limited within the range of [0, 2π]. When the phase function exceeds 2π, it should be processed periodically. Therefore, its phase distribution function can be written as follows:(23)$$\varphi \left(x,y\right)={\mathrm{mod}}_{2\pi }\left[-\frac{\pi \left(x^{2}+y^{2}\right)}{\lambda f}\right]$$

Among them, mod_2π_ is to take the remainder of the phase with respect to 2π. When the focal length meets the limit, the binary Fresnel lens image with the specified focal length can be obtained by programming through Equation (23).

### 2.3. Analysis of Wavelength Effect of Incident Light

From the above content, it can be seen that the maximum beam deflection angle of the LC-SLM and the minimum focal length of the simulated binary Fresnel lens are both affected by the wavelength of the incident light. The larger the wavelength of the incident light, the larger the maximum beam deflection angle and the smaller the minimum focal length, indicating stronger beam deflection ability and focusing characteristics. However, in the actual experimental process, due to the sensitivity of LC-SLM to the wavelength of incident light, if the wavelength of incident light deviates from its working range, its modulation efficiency will be greatly reduced. Using the light of different wavelengths to test the LC-SLM used in this study, it was found that its working range is around 635 nm, and experiments with other wavelengths of light sources cannot obtain modulated light fields. Therefore, LC-SLM plays a decisive role in the selection of the incident light wavelength.

## 3. Experimental Methods

The experimental setup for 3D focusing in confocal microscopy based on an LC-SLM primarily consists of four components: the first is the input light source module, the second is the LC-SLM-based light field modulation module, the third is the light collection and detection module, and the fourth is the data acquisition and system control module. The overall system is illustrated in Figure 4.

The overall optical path can be summarized as follows: the laser beam is collimated and expanded before illuminating the surface of the LC-SLM. After modulation, a focused beam is generated, which passes through the microscope objective and projects a spot onto the sample surface. The light reflected from the sample, containing sample information, is then directed into the data acquisition path and captured by the camera, resulting in an image of the spot with sample information.

The optical path distance between the LC-SLM liquid crystal surface and the microscope objective mount ranges from 320 mm to 400 mm. To facilitate the adjustment of the microscope objective position, the axial focusing parameter of the LC-SLM is set to 350 mm, and the focal length of the phase grayscale map is also set to 350 mm. The collection lens assembly is adjusted to maximize the brightness and clarity of the captured spot.

Taking the spot images obtained during the alignment process as an example, the spot area is smallest at the focal point, and its color is closest to white (255, 255, 255). As the defocus amount increases, the spot area gradually expands, and its color shifts from white to yellow and then to red, indicating a gradual decrease in light intensity. Therefore, a coverage circle of the same size can be used as the sampling region, with its center coinciding with the center of the spot. The sum of the brightness values of all pixels within this coverage circle can be calculated. The brightness of a pixel can be represented by the grayscale value corresponding to its color. Thus, the sum of the grayscale values of all pixels within the coverage circle can be used as the light intensity value of the spot image. The spot size can be determined by extracting the spot boundary from the image and calculating its range, which is then converted into a spot radius value. The spot position can be directly obtained by taking the coordinates of the spot’s central pixel as its location.

In summary, to obtain the characteristic metrics of the spot, the first step is to determine the center point of the spot. Using this point as the center, a circle is drawn with a specific radius, and the sum of the grayscale values of all pixels covered by this circle is calculated. By computing and recording the light intensity values for each 3D focusing point, the relationship between the light intensity of the measured sample and its axial position can be obtained. Based on the above approach, a MATLAB R2016b program was developed to serve as the foundation for experimental data processing.

## 4. Results

### 4.1. Lateral Focusing Experiment

#### 4.1.1. Experimental Data Acquisition

The LC-SLM used in the experiment has a resolution of 1280 × 960. Therefore, theoretically, the phase grayscale map can be controlled with 1280 units in the x-direction and 960 units in the y-direction. Since the control method is identical in both the x- and y-directions, the experiment was conducted in the direction with a larger number of units, i.e., the x-direction. The displacement platform was adjusted to perform lateral scanning experiments with an axial interval of 2 mm between reflective surfaces. The step distance was set to 10 unit pixels, and the phase grayscale map’s center point was assigned coordinates of (0, 0). The phase grayscale map’s coordinates were gradually shifted from (−640, 0) to (640, 0), and the spot images were captured and processed accordingly.

Some of the spot images obtained from the experiment are shown in Figure 5. As can be seen from the figure, when the offset of the phase grayscale map gradually changes from −640 to 640, the spot position also shifts from the right side to the left side of the image. The trend of spot changes is consistent at different defocus amounts, and the spot sizes are similar. However, the brightness of the background red light varies significantly, with spots at smaller defocus amounts exhibiting higher overall brightness and those at larger defocus amounts showing lower overall brightness. Notably, when the phase grayscale map offset is near 0, the spot brightness is significantly higher compared to spots at other offsets.

#### 4.1.2. Experimental Data Analysis

By batch processing all spot images, the light intensity values and center coordinates corresponding to each offset of the pixel grayscale map at different defocus amounts are obtained, as shown in Figure 6. In all subsequent charts, the light intensity values represent the sum of the grayscale values of all pixels within the coverage circle.

After excluding the spot images at both ends that were blocked by the objective aperture, the light intensity values exhibited minimal changes during lateral movement, remaining largely consistent overall. However, a small peak in light intensity was observed near an offset of 40, consistent with the previously mentioned abnormal increase in light intensity.

The x-values showed smooth changes, while the y-values remained essentially constant. Since the lateral focusing experiment involved altering the position in the x-direction, the x-coordinate was expected to vary with changes in the phase grayscale map, while the y-coordinate was expected to remain unchanged. As indicated by the fitted curves, the x-coordinate changes for both sets of spot images closely followed a linear function, with R^2^ values exceeding 0.99, demonstrating excellent linearity. Meanwhile, the y-coordinate showed almost no variation, resembling a horizontal line, indicating that its value remained nearly constant regardless of the independent variable (offset), which aligns with the experimental expectations.

#### 4.1.3. Analysis of Abnormal Peaks

To investigate the cause of the light intensity peak near an offset of 40, the spot images were resampled for phase grayscale map offsets ranging from −200 to 200. While maintaining the original sampling interval of 10 pixels, the sampling interval was increased to 4 pixels between −50 and 150 for more detailed analysis, and changes in the spot were observed. By moving the phase grayscale map to the abnormal position and gradually shifting the measured plane away from the focal point of the microscope objective, it was found that a central bright spot remained in the image even after the modulated spot disappeared. This confirmed that the source of the central bright spot was unrelated to the measured plane. Through adjustments to the optical path, it was determined that the interfering bright spot originated from reflections on the internal lens surfaces of the microscope objective. Adjusting the angle of the objective mount could eliminate the interfering spot; however, excessive deviation from the optical axis would compromise the integrity of the final imaging. Therefore, the objective mount angle was slightly adjusted to reduce the brightness of the interfering spot below that of the dark red background. This approach effectively eliminated the impact of the interfering spot while minimally affecting the final imaging quality. Subsequent sampling of the spot that yielded the data is shown in Figure 7.

From the light intensity values of the spots, it is evident that after eliminating the interfering bright spot caused by reflections on the microscope objective lens surfaces, the abnormal light intensity peak also disappeared, and the spot light intensity values remained largely consistent. The light intensity values at the same offsets in both measurements showed high consistency, and the trends in light intensity changes were also similar, indicating that this lateral modulation method has stable repeatability. In this dataset, the light intensity values exhibited a slight increasing trend as the offset gradually increased. This may be due to the reflective plane, reinstalled after adjusting the optical path, not being perfectly perpendicular to the optical axis, resulting in a slight tilt of the reflective plane relative to the measurement plane. Consequently, the processed light intensity values represent an inclined plane.

#### 4.1.4. Experimental Error Analysis

Under ideal conditions, the light intensity values of the spots should remain consistent across different offsets. However, the actual measured light intensity values exhibit fluctuations within a certain range. Under the condition that the measured surface remains unchanged, the spot images in the sampling interface are largely consistent. However, after lateral modulation, flickering and jumping of the spot’s bright point can be observed. This indicates that during lateral movement, both the shape of the spot and the grayscale values at each pixel point change; however, the final calculated light intensity values remain relatively stable. Based on these observations, the following factors are speculated to cause interference:Laser Source: The laser used in this experiment is a small LED laser. Compared to the helium-neon lasers commonly used in laboratories, its light source is a square LED, which is focused by a lens at the laser head to form a quasi-point-like beam at a distant point. The beam quality is relatively poor, and its stability is slightly weaker. During prolonged operation, fluctuations in output power and frequency may occur, affecting the shape and brightness of the final spot;Optical Path Structure: The laser emits a Gaussian beam, where the center brightness is higher than the edges. Due to limitations in the experimental setup, no beam shaping was performed on the Gaussian beam. As a result, the light received by the LC-SLM has non-uniform brightness across its surface. During lateral modulation, the modulation region on the liquid crystal surface shifts, causing changes in the brightness of the focused spot, which, in turn, affects the final light intensity calculations;Measured Surface: Due to the influence of the polarizing beam splitter, a mirror could not be used as the measured surface. Instead, a white plastic block with high surface reflectivity was used as the measured object. While the surface appears smooth and flat to the naked eye, at the micro-nano scale, it may be relatively rough, affecting the light intensity values at each lateral modulation position. Additionally, if the measured surface is not perfectly perpendicular to the optical axis during installation, it can also lead to variations in the final measured light intensity values.

### 4.2. Axial Focusing Experiment

#### 4.2.1. Experimental Data Acquisition

First, the axial position of the reflective plane was adjusted in steps of 50 μm, moving it from a position with a large defocus amount to one with a small defocus amount. The spot images were observed, and the position where the central spot exhibited a clear outline and could be fully distinguished from the background was identified as the focusing range. A subset of spot images was extracted at equal intervals from the experimental data and arranged into a matrix to illustrate the trend of spot changes. The resulting trend is shown in Figure 8.

From the images, the spot images meeting the sampling criteria can be roughly divided into three categories based on their characteristics.

The first category includes images with earlier numbers (large defocus amounts). Their background light appears dark red and unevenly distributed, with a small central spot area. The second category includes images with middle numbers (medium defocus amounts). Their background light appears bright red, relatively full, and evenly distributed, with a larger central spot area. The third category includes images with later numbers (small defocus amounts). Their background light appears bright red and evenly distributed, with an extremely large central spot area and discrete yellow bright points around the central spot. Within the adjacent range of the second and third categories, the central spot area is smaller, the brightness changes significantly, and the background light remains largely unchanged without being affected by discrete bright points. Therefore, measurements within this range make it easier to distinguish different axial positions, resulting in higher accuracy and precision.

To verify this hypothesis, the step distance for axial movement was reduced to 1 μm for sampling within this range. Based on data provided by the displacement platform control program, the focal position of the microscope objective in this experiment was set to 28.5 mm. Sampling was performed over a 1 mm distance within the adjacent range of the second and third categories, with a step size of 1 μm. The sampling range was from 23.75 mm to 24.75 mm, i.e., 3.75 mm away from the focal point.

#### 4.2.2. Experimental Data Analysis

The collected spot light intensity values are plotted as a scatter plot and presented in Figure 9. Since the experimental setup was reconfigured, the data from this experiment cannot be directly compared with that from the previous section. Additionally, with the reduced focusing range, the changes in spot area are less pronounced compared to the previous section. Therefore, the maximum spot radius value was directly used as the coverage circle radius, resulting in a larger coverage circle area in this data processing step and, consequently, higher calculated light intensity values.

From the data, it can be observed that after reducing the modulation range and increasing the sampling frequency, the spot light intensity values exhibit a strong correlation with distance, and the changes in light intensity show high linearity. However, in the [0.90, 1.00] (unit: mm) range, the light intensity distribution is more scattered. The spots in this region primarily exhibit third-category characteristics, indicating that although the light intensity changes significantly for these spots, they exhibit drift, which can cause substantial interference in high-precision measurements. Therefore, the data from the [0.90, 1.00] (unit: mm) range were excluded, and only the data with stable light intensity distributions were retained for fitting. At this point, the focusing range is 3.85 mm from the focal point. The fitting formulas are given in Equations (24) and (25), and the fitting results are shown in Figure 10.(24)$$y_{1}=3.5927\times 10^{-5}\cdot x^{3}-1.7144\times 10^{-2}\cdot x^{2}+46.1430x+121,859,R^{2}=0.9772$$(25)$$y_{2}=56.914x+118,936,R^{2}=0.9603$$

From the images and data, it is evident that after excluding the third-category characteristic images, the light intensity distribution exhibits a clear linear relationship. This demonstrates that the system can achieve axial measurements with a precision exceeding 1 μm. Both the linear fitting function and the polynomial fitting function show a high degree of overlap with the light intensity values. When the polynomial order reaches three, the R^2^ value no longer shows significant changes, with R^2^ = 0.9772. The R^2^ value of the linear fitting function also exceeds 0.96, indicating a high degree of fitting accuracy. By substituting the distance values into the polynomial fitting function, the experimental light intensity values for each spot were calculated, and the distribution of relative errors was analyzed, as shown in Figure 11.

From the frequency distribution histogram, it can be seen that the relative errors of the experimental light intensity values for each spot are mostly distributed within the range of (−3.01%, 2.29%], accounting for 94.45% of the data. Additionally, 81.35% of the experimental light intensity values fall within the relative error range of (−1.95%, 1.23%]. These data indicate that the light intensity values calculated using the fitting function have high accuracy. If linear fitting is used for calculations, the computational load can be significantly reduced while maintaining similar precision, which is advantageous for the miniaturization of subsequent measurement device processing units. According to the polynomial fitting function Equation (24), in this experiment, every 1 μm movement of the reflective surface corresponds to an average change of approximately 60 units in light intensity. This means that, under ideal conditions, each change in the measured light intensity value reflects an average axial distance change of 16.667 nm. Without modifying the binary Fresnel lens, the measurement accuracy has reached a level comparable to that of LC-SLM-based confocal microscopy systems.

#### 4.2.3. Experimental Error Analysis

In the axial modulation experiment, there is still a certain gap between the measured light intensity values and the theoretical values. As shown in Figure 11, during the process of defocus amount changes, the central spot does not scale uniformly with a fixed shape but changes irregularly as the defocus amount varies, exhibiting only an overall trend of increasing or decreasing in size. After reducing the step distance, a small number of spots exhibit light intensity values that deviate from the fitted values, with a maximum difference of 11–12%, and these deviating light intensity values are mostly lower than the fitted values. Through analysis, this discrepancy may be caused by the following factors, leading to some errors in the experimental results:Optical Path Structure: The laser emits a Gaussian beam, where the center brightness is higher than the edges. After passing through the collimating beam expander, it can be approximated as a plane wave; however, in reality, the light intensity distribution is non-uniform. Therefore, the light received by the LC-SLM also has non-uniform brightness across its surface. During axial modulation, the quasi-collimated beam obtained through the microscope objective has components with certain deflection angles. When the axial distance changes, the positions and intensities of these components also change, causing the captured spot images to vary irregularly with the defocus amount, showing only an overall trend in size changes;Alignment Precision: When setting up and aligning the optical system, the installation angles and heights of numerous components, such as the LC-SLM, PBS, microscope objective, and industrial camera, require manual adjustment, which affects the alignment of all subsequent components and devices. The final impact is that the components are not perfectly collimated or perpendicular, which affects the uniformity of the modulated beam in the focusing range, leading to deviations in the spot light intensity values during axial focusing;Ambient Light Interference: All images in this experiment were captured under dark conditions; however, the lights from electronic equipment around the laboratory are unavoidable. Additionally, the LC-SLM modulation, camera capture, and air-bearing platform control during the experiment are all operated via a computer; therefore, the confocal microscopy 3D focusing system is exposed to interference light from the display. The display screen changes during the experiment, causing the brightness of the interference light to vary accordingly. Although changes in pixel grayscale values are not visible to the naked eye, they may still interfere with the data acquisition process.

These factors contribute to the differences between the measured light intensity values and the fitted values. However, errors caused by alignment precision and ambient light interference are difficult to avoid. Therefore, to maximize the accuracy of the experimental results, it is necessary to standardize the experimental procedures and minimize any negative impact on the results.

## 5. Conclusions

This paper primarily investigates the confocal microscopy 3D focusing measurement method based on LC-SLM and conducts experiments on the 3D focusing characteristics of binary Fresnel lenses. The spots modulated by the binary Fresnel lens exhibit excellent stability in lateral movement. When the axial modulation range is within 900 μm at a distance of 3.85 mm from the objective focal point, axial measurements with a precision exceeding 1 μm can be achieved. Under ideal conditions, the limit resolution can reach 16.667 nm, and a linear function expression can be used for calculations, significantly simplifying the computational load with minimal impact on measurement accuracy. During the 3D focusing process, the system does not recalculate or regenerate the phase grayscale map of the binary Fresnel lens but can rapidly change the lateral position of the spot, achieving axial measurements with precision comparable to LC-SLM-based confocal microscopy systems.

## Figures and Tables

**Figure 1 sensors-25-02620-f001:**
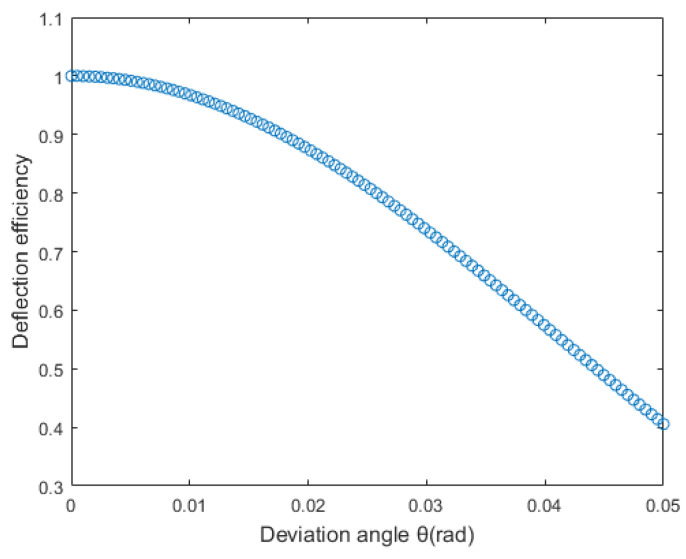
Beam deflection efficiency curve.

**Figure 2 sensors-25-02620-f002:**
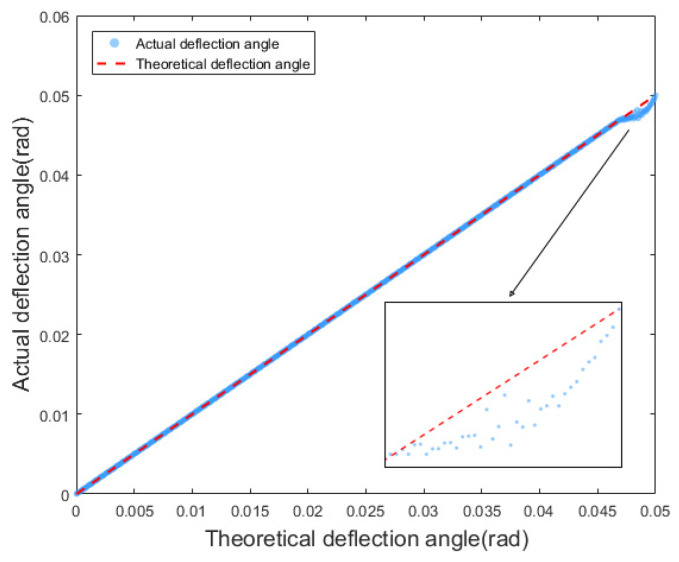
Simulation curve of actual beam deflection angle.

**Figure 3 sensors-25-02620-f003:**
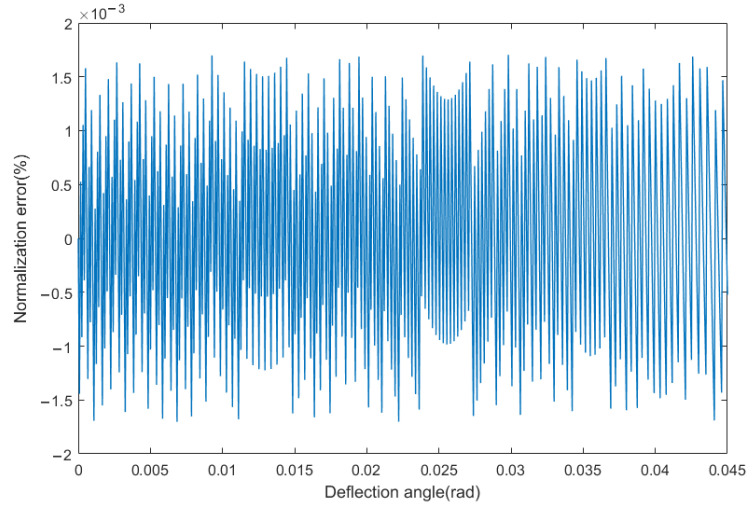
Normalized error curve for *M* = 256.

**Figure 4 sensors-25-02620-f004:**
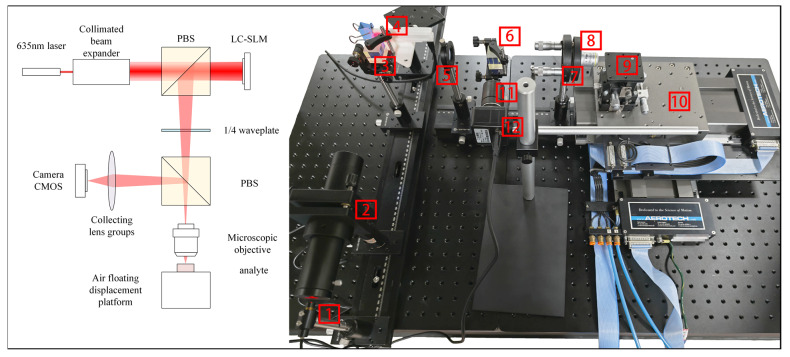
Three-dimensional focusing experimental system setup: 1. 635 nm laser; 2. collimating beam expander; 3. polarizing beam splitter (PBS); 4. LC-SLM; 5. quarter-wave plate (1/4 Wave Plate); 6. polarizing beam splitter (PBS); 7. objective mount; 8. microscope objective; 9. three-axis translation stage; 10. high-precision air-bearing translation stage; 11. collection lens assembly; 12. camera.

**Figure 5 sensors-25-02620-f005:**
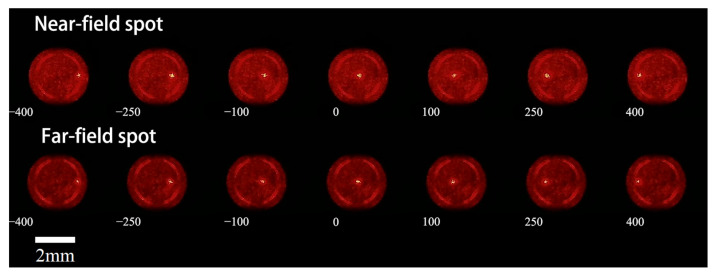
Spot lateral movement image: the number below the spot image represents the offset (in pixels) of the current phase grayscale map.

**Figure 6 sensors-25-02620-f006:**
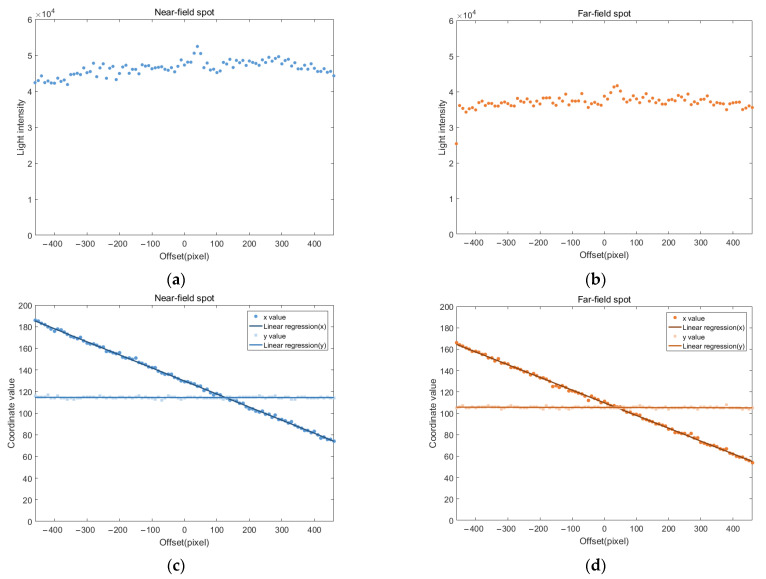
Lateral focusing experimental results: (**a**) light intensity distribution of the near-field spot; (**b**) light intensity distribution of the far-field spot; (**c**) position variation curve of the near-field spot; (**d**) position variation curve of the far-field spot.

**Figure 7 sensors-25-02620-f007:**
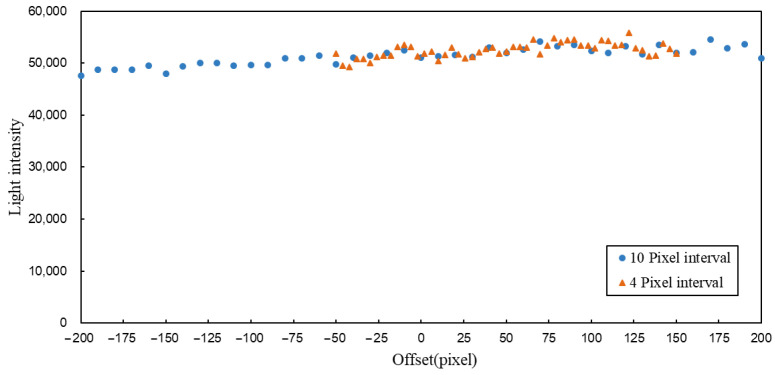
Scatter plot of light intensity distribution.

**Figure 8 sensors-25-02620-f008:**
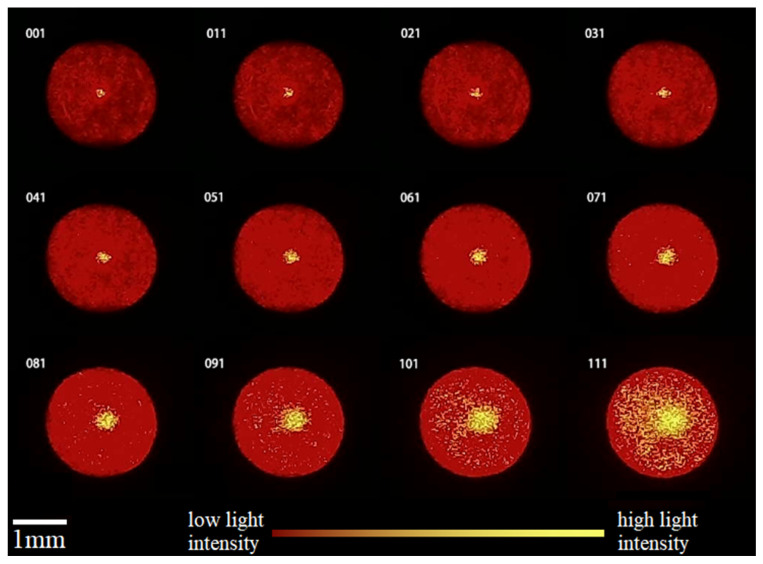
Trend of axial focusing spot changes.

**Figure 9 sensors-25-02620-f009:**
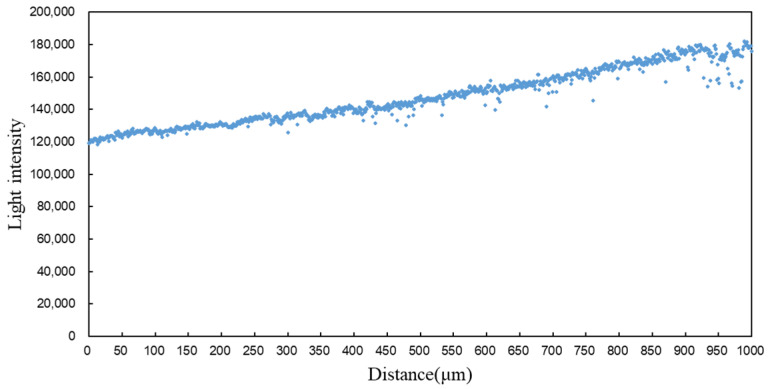
Light intensity distribution of spots in the adjacent range of the second and third categories.

**Figure 10 sensors-25-02620-f010:**
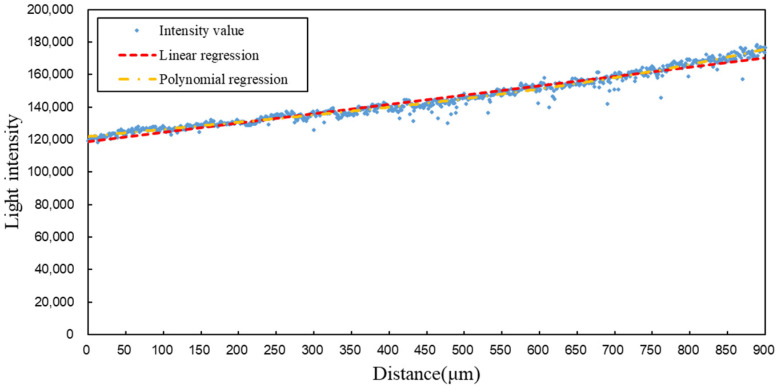
Light intensity distribution and fitting curve after excluding third-category characteristic images.

**Figure 11 sensors-25-02620-f011:**
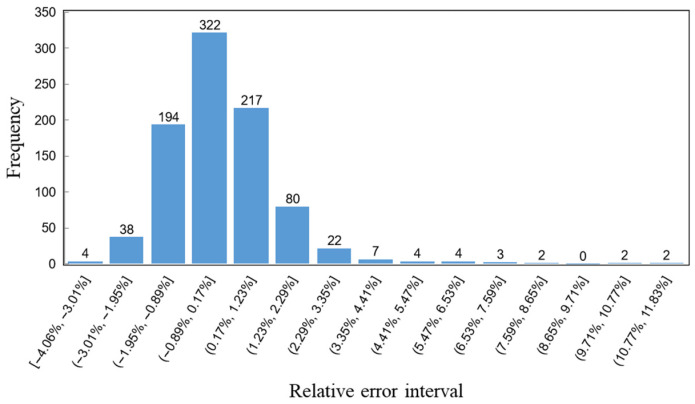
Histogram of relative error frequency distribution.

## Data Availability

The original contributions presented in this study are included in the article. Further inquiries can be directed to the corresponding author.

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
