# Peer review of "Three-Dimensional Focusing Measurement Method for Confocal Microscopy Based on Liquid Crystal Spatial Light Modulator"

_sensors, 2025, doi:10.3390/s25082620_

Round 1
Reviewer 1 Report
Comments and Suggestions for Authors
In this manuscript, the authors propose a 3D focusing strategy for Confocal Microscopy Based on Liquid Crystal Spatial Light Modulator. This method allows rapid adjustment of lateral focusing positions without regenerating phase grayscale maps, achieves comparable axial measurement accuracy, and enhances measurement speed. I suggest to accept this manuscript in its current form.
Author Response
Comments 1: I suggest to accept this manuscript in its current form.
Response1 : We are deeply grateful to Reviewer 1 for the supportive assessment and valuable endorsement of our work. Your recognition of the manuscript's readiness for publication is particularly encouraging.
Reviewer 2 Report
Comments and Suggestions for Authors
The authors of the reviewed work proposed a confocal measurement method using a liquid crystal spatial light modulator (LC-SLM) to simulate a binary Fresnel lens for 3-D focusing, enabling non-mechanical measurement of spatial positions on sample surfaces. The 3-D focusing method based on LC-SLM was presented in detail, a 3D confocal microscope focusing system was constructed, and lateral and axial focusing experiments were conducted.
The obtained experimental results show that the system can freely adjust the lateral focusing positions. In the axial focusing range of 900 μm, it achieves an axial measurement accuracy exceeding 1 μm, with a maximum resolution of about 16.667 nm. In the theoretical introduction, the modulation characteristics of the LC-SLM lenses used in the experiment were analyzed and a method for constructing binary Fresnel lenses was proposed. Then, the 3-D Focusing Experimental System Setup is presented and described in detail.
The obtained results are presented in graphs and photos, the quality of which is sufficient. The conclusions describe the basic results obtained in the work.
The cited literature, although quite modest, enables verification and comparison of the obtained results.
Many of the relationships presented in the graphs show a linear relationship. Polynomial regression is also used. What information can be obtained from the fit coefficients of such functions?
The work may be published in the journal Sensors.
Author Response
Comments 1: Many of the relationships presented in the graphs show a linear relationship. Polynomial regression is also used. What information can be obtained from the fit coefficients of such functions?
Response 1: The data demonstrating linear relationships in the article are primarily shown in Figs. 6(c) and (d), while the coefficients and their corresponding fitting functions are explicitly provided in Eqs. 24 and 25. Regarding Figs. 6(c) and (d): Although the numerical values of the fitting coefficients are not listed in the text, the figures clearly indicate that better linearity of the fitting function correlates with smoother transverse movement and higher precision in displacement of the light spot. Regarding Eqs. 24 and 25: The fitting coefficients in these equations reflect the relationship between the axial position of the measured surface and light intensity values. The specific values of these coefficients depend on the methodology for calculating light intensity, such as the selection of the coverage circle radius.
Reviewer 3 Report
Comments and Suggestions for Authors
Please see attached review report.
